# A Two-Point Ultrasound-Guided Injection Technique for the Transversus Thoracis Plane Block: A Canine Cadaveric Study

**DOI:** 10.3390/ani12172165

**Published:** 2022-08-24

**Authors:** Manuel Alaman, Cristina Bonastre, Adrián González-Marrón, Ekaterina Gámez Maidanskaia, Alicia Laborda

**Affiliations:** 1Department of Animal Pathology, Faculty of Veterinary Medicine, University of Zaragoza, C/Miguel Servet 177, 50013 Zaragoza, Spain; 2Hospital Veterinario Anicura Valencia Sur, Avda. Picassent, 28, 46460 Silla, Spain; 3Instituto Universitario de Investigación Mixto Agroalimentario de Aragón (IA2), University of Zaragoza, C/Miguel Servet 177, 50013 Zaragoza, Spain; 4Group of Evaluation of Health Determinants and Health Policies, Department of Basic Sciences, Universitat Internacional de Catalunya, 08159 Barcelona, Spain; 5Anaesthesiology and Pain Therapy Division, Department of Clinical Veterinary Medicine, Vetsuisse Faculty, University of Bern, 3012 Bern, Switzerland

**Keywords:** dog, intercostal nerve, mastectomy, parasternal block, regional anaesthesia, sternotomy

## Abstract

**Simple Summary:**

The transversus thoracis plane block is a locoregional technique recently described in canine cadavers to desensitize the intercostal nerves running through this plane. In canine cadavers, a transverse approach through a single injection point at the fifth intercostal space has been described, although consistent staining of the intercostal nerves was not completely achieved. The objective of this study was twofold: (1) to evaluate if the transverse approach is feasible at the third and sixth intercostal spaces and (2) to compare, by anatomical dissection, the spread of a dye solution and the staining of the intercostal nerves when a low volume (0.5 mL kg^−1^) or a high volume (1 mL kg^−1^) was equally divided at a two-point injection in the same hemithorax. Our results showed that the injection of the high-volume dye solution, equally injected at the third and sixth intercostal spaces using the transverse approach, achieved a consistent staining of from T2 to T7 intercostal nerves. This block could achieve adequate desensitization of the ventral chest wall during sternotomy in the dog. Clinical studies in live animals are necessary to confirm the efficacy of this technique.

**Abstract:**

The transversus thoracis plane (TTP) block desensitizes the intercostal nerves that run through this plane, providing analgesia to the ventral thoracic wall. Two canine cadavers were used to assess the feasibility of the transverse approach for the TTP (t-TTP) under ultrasound guidance to inject a solution at the third and sixth intercostal spaces. Eight cadavers were used to compare the spread and number of intercostal nerves that were stained when a low volume (LV) 0.5 mL kg^−1^ or a high volume (HV) 1 mL kg^−1^ of a dye-lidocaine solution was injected into the same hemithorax, injecting the volume equally at these intercostal spaces using the transverse approach. Fisher’s exact test and Wilcoxon signed-rank test were used to contrast the spread of the different volume solutions. The injectate spread along the TTP after all injections, dying a median number (range) of 3 (2–5) and 6 (5–6) nerves with LV and HV, respectively (*p* = 0.011). The two-point injection of HV, using the t-TTP approach, is a feasible technique that provides a consistent staining from T2 to T7 intercostal nerves. The injection of HV instead of LV increases the spread and enhances the number of stained intercostal nerves.

## 1. Introduction

Several anaesthetic blocks have been described to provide analgesia for the ventral thoracic wall in human medicine [1]. These techniques could control the nociceptive stimulus related to thoracic surgery and reduce the perioperative opioid consumption and its side effects [2,3]. The use of opioids has been associated with decreased pulmonary functional residual capacity, respiratory depression and hypoxaemia [4]. Thus, the use of locoregional anaesthetic techniques could be recommended in thoracic surgery as a part of a balanced anaesthetic approach [5].

In veterinary medicine, thoracic epidural [6], thoracic paravertebral block [7,8,9,10,11], erector spinae plane block [12,13,14], intercostal blocks [6,15], transversus thoracis plane (TTP) block [16,17] and pecto-intercostal block [18] could be useful, providing analgesia to the ventral chest wall.

The TTP block is performed under ultrasound guidance injecting the anaesthetic solution in the fascial plane between the transversus thoracis (TT) and the internal intercostal muscles. In human medicine, this anaesthetic block desensitizes the anterior portion of the intercostal nerves, which innervate the thoracic ventral midline skin, sternum and pleura [19,20]. The analgesia provided by the TTP block has been effective in humans undergoing pericardiocentesis [21], median sternotomies [21,22,23,24] or breast surgery [25,26] or to alleviate chronic pain related to sternal injuries [27,28]. Bilateral catheter placement has been reported for continuous TTP block in a human patient undergoing median sternotomy [29].

In dogs, sagittal [16] and transverse (t-TTP) [17] approaches for TTP block have recently been described in cadaveric studies. Although both approaches have only been reported in canine cadavers, they could enable the distal injection of an anaesthetic solution into the TTP in live dogs and block from the second to seventh intercostal nerves (T2-T7), which run through the TTP and innervate the sternum and adjacent tissues. In a previous study, the authors reported the spread of a methylene blue and lidocaine solution through the TTP and the dying of the intercostal nerves that run along this plane when it was injected into the TTP through a single injection point at the fifth intercostal space [17]. Although they observed that the spread and number of stained nerves were greater when a larger volume was administered, the higher volume (1 mL kg^−1^) did not achieve the staining of all intercostal nerves running along the TTP.

Thus, the objectives of this study were: (1) to assess if the ultrasound-guided t-TTP approach can recognise the TTP at the third and sixth intercostal spaces and to inject a solution into this plane; (2) to compare the spread of a low-volume 0.5 mL kg^−1^ (LV) and a high-volume 1 mL kg^−1^ (HV) methylene blue and lidocaine solution when each was injected using the t-TTP approach at the same hemithorax and equally divided in two injections at the third and sixth intercostal spaces.

We hypothesized that: (1) the transversus approach to the TTP can ultrasonographically recognise the TTP at the third and sixth intercostal spaces and inject a solution into this plane. (2) A methylene blue and lidocaine solution injected in the TTP, using a t-TTP approach at the third and sixth intercostal spaces, will stain the intercostal nerves that run through the TTP; more intercostal nerves will be stained when injecting HV solution versus LV.

## 2. Materials and Methods

### 2.1. Study Design

This study received the approval of the Animal Ethics Experimentation Committee of the University of Zaragoza (PI43/22). A total of 10 canine cadavers were included in the project. Exclusion criteria included the presence of congenital sternal malformations and/or penetrating abdominal or thoracic injuries. All dogs were euthanized for reasons unrelated to the study and immediately frozen in individual black bags. Cadavers were thawed at room temperature 72 h before the study.

The study was divided into two phases. In the first phase, two cadavers were used to assess if the t-TTP approach could recognise the TTP at the third and sixth intercostal spaces and inject a solution into this plane. In the second phase, a total of 8 cadavers were used to evaluate the spread of a methylene blue and lidocaine solution and the staining of the intercostal nerves when LV or HV of this solution was injected into the same hemithorax, dividing the injected volume equally between the third and sixth intercostal spaces. In the second phase, the decision to include 8 canine cadavers was based on the sample size used in similar cadaver studies. The anatomical evaluation and all the injections were performed ultrasound-guided using a linear ultrasound transducer (14–6 MHz, L14-6Ns, Mindray Bio-Medical Electronics Co., Shenzhen, China) and a portable ultrasound machine (Mindray M9 Vet, Mindray Bio-Medical Electronics Co., Shenzhen, China). This phase was performed by the same experienced researcher (MA).

### 2.2. Phase I: Two-Point T-TTP Injection

Among the available dogs, two were randomly selected and their thoracolumbar region was clipped. Each cadaver was positioned in dorsal recumbence with cranially oriented forelimbs. A dissection was carried out in only one hemithorax of each of the two cadavers. After removing the skin, the pectoralis superficialis and pectoralis profunda muscles were separated from the sternum and laterally reclined. The rectus abdominis muscle, which was sectioned between the first costal cartilage and caudal to the xiphoid process, as well as the rectus thoracis muscle, were reclined. The external and internal intercostal muscles present between the third and fourth and the sixth and seventh costal cartilages were separated from the sternum and their caudal costal cartilage insertion and cranially reclined to visualize the TTP. The internal intercostal membrane, the path of the intercostal nerves, the internal thoracic artery and vein and the TT muscle, were identified in both intercostal spaces. Subsequently, the TT muscle was separated from the endothoracic fascia and costal pleura. After that, the other hemithorax was ultrasonographically evaluated along the regions between the third and fourth and the sixth and seventh costal cartilages. The transducer was positioned on the third and the sixth intercostal spaces in a parasternal position, parallel to the longitudinal axis of the costal cartilages and slightly lateral to the sternebra, as the authors described in a previous study of the t-TTP approach [17]. The sonoanatomy of both regions was correlated with the gross anatomical findings observed during dissections to recognize the references to perform the ultrasound-guided t-TTP. Gross and ultrasonographic anatomical differences between the third and sixth intercostal spaces were recorded. Once the investigators became familiar with the sonoanatomy, 0.1 mL of methylene blue (Methylthionium Chloride Injection 1% *w*/*v*; Martindale Pharma, Brentwood, UK) was injected using the t-TTP at the third and sixth intercostal spaces, laterally to the internal thoracic artery and vein, using a 63 mm × 22 G spinal needle (BD Spinal Needle; BD Medical, Franklin Lakes, NJ, USA). Finally, the dissection procedure was repeated to confirm the spread of the dye into the TTP.

### 2.3. Phase II: Evaluation of the Injectable Solution Spread

The injectable solution consisted of a mixture of 200 mL lidocaine (lidocaine 2%, B. Braun Melsungen AG, Melsungen, Germany) and 0.2 mL of methylene blue. For the HV and LV injections, the total volume was equally divided into two syringes, with 0.5 mL kg^−1^ and 0.25 mL kg^−1^ of the injectable solution for each of the eight cadavers, respectively. The HV injections were performed into the TTP at the same hemithorax at the third and sixth intercostal spaces. The LV injections were performed at the contralateral hemithorax using the same procedure. Cadavers were randomized (Microsoft Excel for Windows Version 2013, Microsoft Corporation, Redmond, WA, USA) as to which hemithorax, right or left, and which volume of the solution, LV or HV, was injected. After all injections were performed in the same cadaver, the spread of the injectate and the staining of the intercostal nerves were evaluated by anatomical dissection. The injections and dissections of the cadavers were performed by different researchers (MA and CB, respectively), with second investigator blinded to the volume of solution injected at each hemithorax.

#### 2.3.1. Ultrasound-Guided T-TTP Injection

Each cadaver was placed in dorsal recumbency with forelimbs cranially oriented and its thoracolumbar region clipped. The linear ultrasound transducer was placed in a parasternal position at the third intercostal space, slightly lateral to the sternum and parallel to the longitudinal axis of the adjacent costal cartilages with a medially oriented marker (Figure 1). The following thoracic structures were identified: pectoralis profunda, rectus abdominis and external and internal intercostal muscles, internal intercostal membrane, third sternebra, internal thoracic artery and vein, transversus thoracis muscle and costal pleura (Figure 2).

A 63 mm × 22 G spinal needle (BD Spinal Needle; BD Medical, Franklin Lakes, NJ, USA), was inserted in a ventro-medial-to-dorso-lateral direction using an in-plane technique. The needle was advanced through the pectoralis profunda, rectus abdominis, intercostal muscles and internal intercostal membrane until the tip was positioned in the TTP. The target plane was recognized between the transversus thoracis muscle and the internal intercostal membrane and muscle, lateral to the internal thoracic artery and vein (Figure 2). The correct positioning of the needle tip was confirmed by injecting 0.5 mL of saline and visualising the formation of a pocket of fluid into the TTP. If the injectate did not reach the target plane, the needle tip was redirected. Then, one of the HV or LV syringes (with 0.5 mL kg^−1^ for the HV treatment and 0.25 mL kg^−1^ for the LV) was injected at the third intercostal space, based on randomization assignments. The procedure was repeated at the sixth intercostal space of the same hemithorax injecting the same volume (Figure 3). On the contralateral hemithorax, the procedure was performed in the same way but injecting the other study volume. After all injections, the quality of needle tip ultrasound visualization was scored as absent, poor or good (Appendix A Table A1).

#### 2.3.2. Anatomical Dissection

The dissection of both hemithorax began with an incision along the ventral midline skin. The pectoralis superficialis and pectoralis profunda muscles were separated from their sternal attachments and laterally reclined. Rectus thoracis and rectus abdominis muscles were sectioned at the first costal cartilage and caudally to the xiphoid process and reclined dorsally. Then, the thoracic wall was bilaterally and longitudinally sectioned to its sagittal midline. The cervical and abdominal muscles that attach to the ventral chest wall, the diaphragm, the ventral mediastinum, and the internal thoracic arteries and veins were sectioned before they go through the TTP. The ventral chest wall was removed to expose its inner aspects. Costal pleura and TT muscles were bilaterally sectioned along the longitudinal axis of the TT muscle, exposing the TTP and T2 to T7 intercostal nerves. The nerves with all their quadrants dyed more than 1 centimetre in length were considered to be stained.

Five different locations were evaluated to record the spread pattern of the injectable solution: (1) the plane between the rectus abdominis muscle and the intercostal muscles; (2) the transversus thoracis plane (3) the plane between the TT muscle, the endothoracic fascia and the costal pleura; (4) the mediastinum; (5) the intrapleural space. TTP was divided in segments, taking each intercostal space as a reference. Segments were considered dyed when staining was observed along their longitudinal axis. 

### 2.4. Statistical Analysis

IBM SPSS statistics version 19.0 for Windows (SPSS Inc., Chicago, IL, USA) was used to perform statistical analysis. The quantitative variables were described with median and range and qualitative variables with absolute frequency and percentage. The spread of the injectable solution along the five locations described above and the staining of the intercostal nerves were compared based on the volume that was injected (HV and LV). Wilcoxon signed-rank tests were used to compare the mean ranks of TTP intercostal segments and intercostal nerves stained with HV and LV methylene blue and lidocaine solution. The proportions of intercostal nerves and TTP segments stained according to the volume injected were evaluated using the Fisher´s exact test. The significance level was set at *p* < 0.05.

## 3. Results

Two cadavers, weighing 8.2 and 20.3 kg, were used in phase I, and eight dogs, with a mean weight of 8.4 kg (SD ± 7.1), were used in phase II.

### 3.1. Phase I: Two-Point T-TTP Injection

The sternebrae corresponding to the third and sixth intercostal spaces were visualized as a triangular structure with a hyperechoic surface and an acoustic shadow underneath. In both intercostal spaces, the pectoralis profunda muscle was visualized as a thick hypoechoic structure located lateral and ventral to the sternum. The rectus abdominis muscle, located between the pectoralis profunda muscle and the intercostal muscles, was observed as a hypoechoic structure, with a thin parasternal hyperechoic line at the level of the sixth intercostal space, and only as a thin hyperechoic line at the third space. In both spaces, the external intercostal muscle was observed as a hypoechoic structure covering the lateral portion of the internal intercostal muscle, except for its most proximal aspect to the sternum. The internal intercostal muscle appeared as a hypoechoic structure, extending in both spaces to the lateral aspect of the corresponding sternebra. The internal intercostal membrane was observed in both spaces as a hyperechoic line that was intimately related to the dorsal aspect of the internal intercostal muscles. The TT muscle, observed as a hypoechoic structure, was located dorsal to the intercostal muscles and ventral to the costal pleura, which was identified as a thin hyperechoic line. At the sixth intercostal space, the TT muscle extended slightly more laterally than at the third space. At both intercostal spaces, the TTP was visualised as a virtual space, delimited by the internal intercostal membrane and the TT. TTP contained a variable amount of fat and also the internal thoracic artery and vein, both visualized as two rounded anechoic structures next to the sternum.

The anatomical landmarks delimiting the TTP could be recognised at the third and sixth intercostal space of the two cadavers used during this phase (Figure 2 and Figure 3). During anatomical dissections, the presence of dye in the TTP was confirmed after all injections.

### 3.2. Phase II: Evaluation of the Injectable Solution Spread

#### 3.2.1. Ultrasound-Guided T-TTP Injection

The TTP ultrasound landmarks could be recognized during all injections. At the third intercostal space, the visualization of the needle tip into the TTP was scored as good in 12 out of 16 injections and poor in 4 out of 16 injections. At the sixth intercostal space, the needle tip visualization was scored as excellent in 12 out of 16 injections, regular in 3 out of 16 and poor in 1 out of 16. In all injections, the visualization of a pocket of fluid delimited by the TT muscle and the internal intercostal membrane was evident.

#### 3.2.2. Evaluation of Methylene Blue and Lidocaine Solution Spread

A median (range) of 3 (2–5) intercostal nerves using LV and 6 (5–6) nerves using HV were stained (*p* = 0.011). In all cadavers, HV injection succeeded in staining more intercostal nerves than LV.

The methylene blue and lidocaine solution was observed along the longitudinal axis of the TTP (Figure 4) staining 4 (3–5) and 6 (5–6) (median (range)) intercostal segments after LV and HV injections, respectively (*p* = 0.016). Moreover, HV and LV injections resulted in staining of the plane between the TT muscle, endothoracic fascia and costal pleura. The presence of injectable solution in the ventral mediastinum was observed in all cadavers. Intrapleural staining was recorded in 1 out of 8 cadavers. In this case, the intrapleural injection was ultrasonographically detected during the execution of the t-TTP approach and the needle tip was repositioned into the TTP.

All the data referring to the detection of the methylene blue-lidocaine solution are shown in Figure 5 and Table 1 and Table 2.

## 4. Discussion

The present study demonstrates that the TTP is an identifiable plane at the third and sixth intercostal spaces, following the sonoanatomical landmarks described by the authors for the t-TTP approach [17]. Furthermore, this two-point approach performed at the third and sixth intercostal spaces enables the injection of a methylene blue and lidocaine solution in the TTP. The ultrasound guidance for t-TTP approach allows for a complete visualisation of the needle and the injectate in real time at both intercostal spaces, improving the accuracy and safety of the technique.

As other studies have reported [16,17], the injection of an anaesthetic–dye solution into the TTP resulted in its cranio-caudal and latero-medial spread along the extension of TTP and the dying of T2 to T7 intercostal nerves. Moreover, in the present study, a significantly higher rate of staining of intercostal TTP segments and intercostal nerves was obtained when HV was injected compared to LV. These results, together with those reported by other studies [16,17], suggest that increasing the injected volume of an anaesthetic–dye TTP solution would produce a more extensive spread of the solution, increasing the probability of desensitizing a higher number of intercostal nerves and providing a more effective analgesia for the ventral chest wall.

The injectable solution composition and the LV and HV injected into each hemithorax were the same as those used in a previous study carried out by the authors [17]. In the current study, these volumes were distributed in two injection points at the third and sixth intercostal spaces of the same hemithorax, whereas, in the previous study, the total volume was injected by a single-point injection at the fifth intercostal space. This single-point injection technique resulted in 3 (2–4) and 4 (3–5) (median (range)) stained intercostal nerves using the LV and HV, respectively, while the two-point injection technique resulted in a median of 3 (range 2–5) for the LV and 6 (range 5–6) for the HV. In addition, the previous study reported a median number of stained TTP intercostal segments of 4 (range 3–5) and 5 (range 4–6) for LV and HV, respectively, while the present study reports a median of 4 (range 3–5) for the LV injection, and 6 (range 5–6) for the HV. The results obtained after the HV two-point injection suggest that this technique would achieve a wider spread of the solution along the TTP and dye more intercostal nerves than the single-point injection technique. However, the spread and the number of stained nerves were similar for both techniques when LV was injected. The authors hypothesise that the injection of small volumes in the LV group (0.25 mL kg^−1^ at each point) could result in incomplete staining of several TTP intercostal segments, which, therefore, would not have been recorded. This incomplete distribution could decrease the probability of staining the intercostal nerves.

In dogs, the distal part of the intercostal nerves from T2 to T7 are involved in the innervation of the ventral chest wall, including the ventral midline skin, the medial aspect of the thoracic mammary tissue, the sternum, the TT, the internal intercostal and rectus abdominis muscles, and the costal pleura [30]. The results of the current study suggest that a two-point injection technique could be effective to desensitize from T2 to T7 intercostal nerves when a HV anaesthetic solution is injected, providing analgesia to the ventral chest wall. Furthermore, this technique would provide a wider and more consistent anaesthetic block than the single-point injection technique when a high volume is administered. However, the injection of 1 mL kg^−1^ local anaesthetic solution could suppose an important clinical limitation, as the therapeutic range of the anaesthetics used could be exceeded or its excessive dilution could be required, with a consequent loss of efficacy. Further cadaveric and clinical studies would be necessary to assess the efficacy of using lower injectable volumes.

In humans, a low rate of complications related to the use of this technique has been reported. [31]. During the present study, staining of the intrapleural space was observed in one cadaver. Although the solution could be injected sonographically and the needle repositioned, errors in the execution of the technique could result in inadvertent cardiac or pulmonary punctures; therefore, attention should always be focused on the path of the needle tip. Although the puncture of the internal thoracic vessels was not noted in any cadaver, the use of colour Doppler in live dogs could improve their identification and reduce the risk of puncture.

In all cases, the dye solution was found in the TTP staining the costal pleura, the TT muscle and ventral mediastinum. As suggested in other studies [17,32], the injectate could reach the mediastinum through the endothoracic fascia, which separates the costal pleura and the TT muscle and enters the ventral mediastinum through the costomediastinal recess. 

Multiple blinded or ultrasound-guided intercostal nerve blocks were commonly used to provide analgesia to patients undergoing median sternotomy [9,15]. Our results and those reported in similar studies [16,17] suggest that the TTP block would reduce the number of injection sites needed to desensitise these nerves. Furthermore, Zublena et al. (2021) suggested that the TTP block would produce less ventilatory involvement than that produced by multiple intercostal blocks, considering that the muscles desensitized by the TTP are not as essential in ventilation as intercostal muscles.

One of the main limitations of this study concerns the use of thawed cadavers, which could produce differences in the spread of the injectable solution compared with alive dogs [33]. Furthermore, the spread of the injectate may be influenced by the movement of the thoracic cage during ventilation, blood flow and lymphatic drainage. Ventilatory movements may also impair the execution of the technique. The ultrasound appearance of the muscles and fascial layers may be altered in cadavers after freezing and thawing, so the appearance of TTP sonoanatomical landmarks may be different in alive animals. The use of methylene blue as the only injected dye may impair the assessment of whether the intrapleural cavity or the ventral mediastinum were stained by the injection of HV or LV. Although the mixture of methylene blue-lidocaine is widely used in other cadaveric studies [34,35], this should be considered another limitation of the study since, the dye’s effects on the physical properties and the spread of the lidocaine are still unknown. 

## 5. Conclusions

The ultrasound-guided t-TTP approach at the third and sixth intercostal spaces is a feasible technique, which could be used to inject a solution into the TTP. Considering the results reported in the current study, the injection of 1 mL kg^−1^ anaesthetic solution, equally injected at the third and sixth intercostal spaces using the t-TTP approach, could produce a consistent block from T2 to T7 intercostal nerves. Despite this, clinical studies are required to assess the efficacy of this high-volume, two-point injection technique to provide analgesia to the ventral chest wall of dogs in vivo, as well as the efficacy of injecting lower volumes to achieve the same purpose.

## Figures and Tables

**Figure 1 animals-12-02165-f001:**
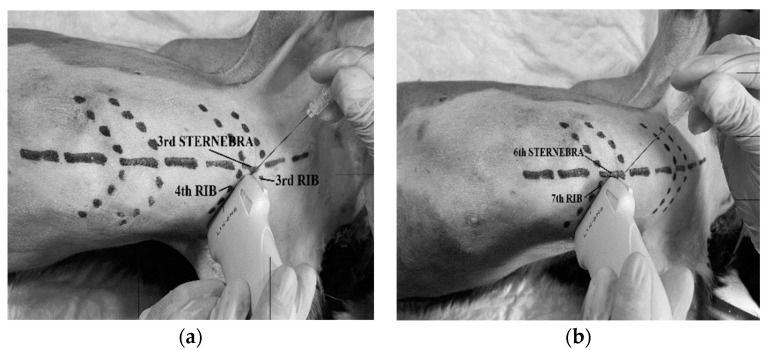
Two-point injection technique for the transverse approach to the transversus thoracis plane under ultrasound guidance. (**a**) The probe was placed parasternal and parallel to the third and fourth ribs, and the marker medially situated and guided from a caudo-lateral to cranio-medial orientation. The spinal needle was inserted from a ventro-medial to dorso-lateral direction using an “in-plane” technique. (**b**), Transducer positioned in the same orientation between the sixth and seventh ribs.

**Figure 2 animals-12-02165-f002:**
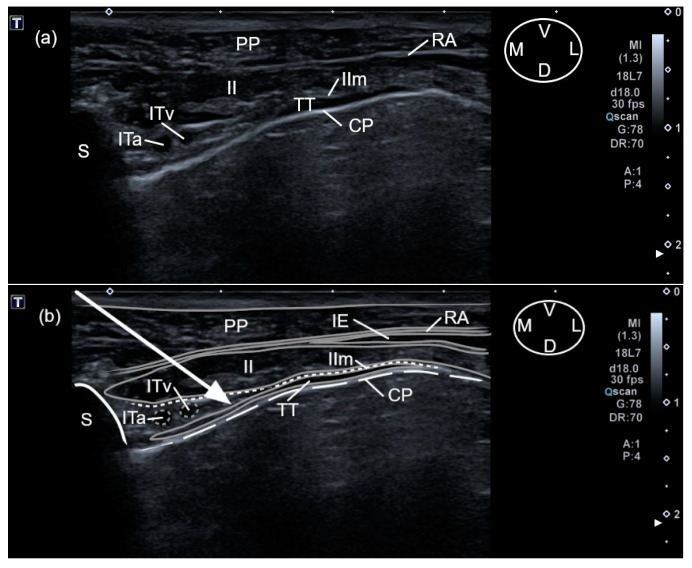
(**a**) Anatomical structures visualized during the two-point ultrasound-guided injection technique for the transverse approach to the transversus thoracis plane at the third intercostal space in a canine cadaver. (**b**) The same visualization of the anatomical structures with the edges of the muscles highlighted to improve the identification (solid grey lines), the internal intercostal membrane (short dashed white lines), third sternebra (solid white line), vascular structures (short dashed grey lines) and costal pleura (long dashed lines). The arrow shows the pathway followed by the needle before reaching the transversus thoracis plane. CP, costal pleura; D, dorsal; IE, external intercostal muscle; II, internal intercostal muscle; IIm, internal intercostal membrane; ITa, internal thoracic artery; ITv, internal thoracic vein; L, lateral; M, medial; PP, pectoralis profunda muscle; RA, rectus abdominis muscle; S, Sternum; TT, transversus thoracis muscle; V, ventral.

**Figure 3 animals-12-02165-f003:**
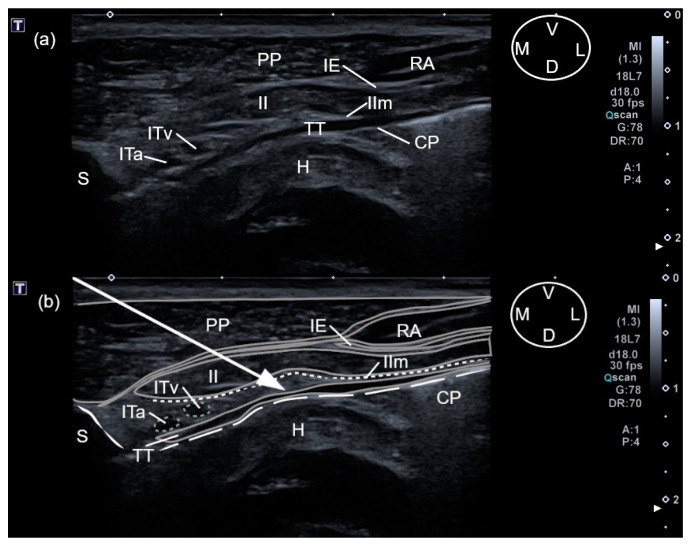
(**a**) Anatomical structures visualized during the two-point ultrasound-guided injection technique for the transverse approach to the transversus thoracis plane at the sixth intercostal space in a canine cadaver. (**b**) The same visualization of the anatomical structures with the edges of the muscles highlighted to improve the identification (solid grey lines), the internal intercostal membrane (short, dashed white lines), sixth sternebra (solid white line), vascular structures (short dashed grey lines) and costal pleura (long dashed lines). The arrow shows the pathway followed by the needle until reaching the transversus thoracis plane. CP, costal pleura; D, dorsal; H, heart; IE, external intercostal muscle; II, internal intercostal muscle; IIm, internal intercostal membrane; ITa, internal thoracic artery; ITv, internal thoracic vein; L, lateral; M, medial; PP, pectoralis profunda muscle; RA, rectus abdominis muscle; S, Sternum; TT, transversus thoracis muscle; V, ventral.

**Figure 4 animals-12-02165-f004:**
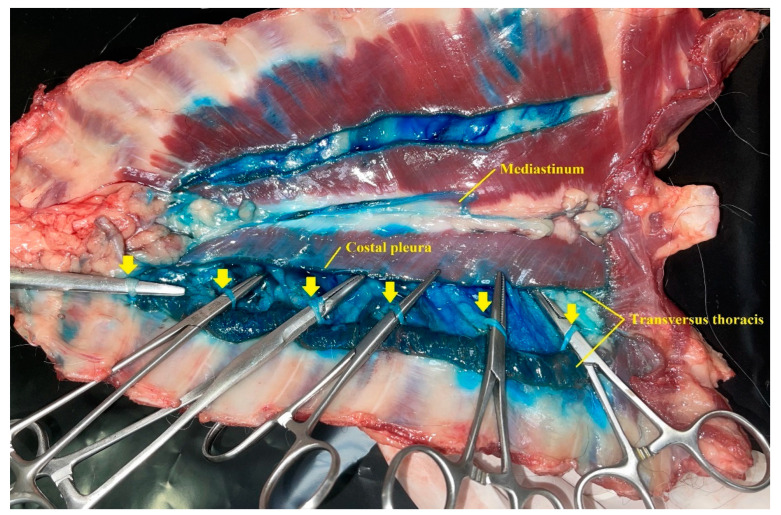
Image of the transversus thoracis plane after performing the blockade and the anatomical dissection. Costal pleura and TT muscle were sectioned along the longitudinal axis of the TT muscle, exposing the TTP and T2 to T7 intercostal nerves (arrows). The methylene blue and lidocaine solution was found in the TTP staining the intercostal nerves and the ventral mediastinum. TT, transversus thoracis; TTP, transversus thoracis plane.

**Figure 5 animals-12-02165-f005:**
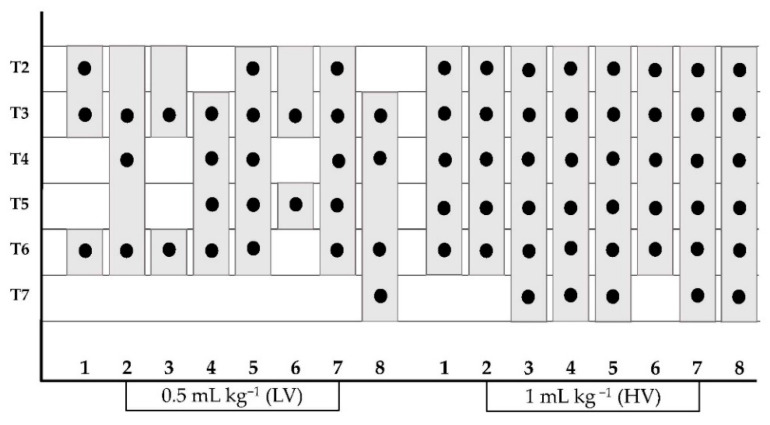
The figure shows the intercostal nerves stained between T2–T7 (black points) and the spread of the injectate along the intercostal TTP segments (grey columns) after the injection of a low volume (0.5 mL kg^−1^) and a high volume (1 mL kg^−1^) of the methylene blue and lidocaine solution in eight canine cadavers (a total of sixteen two-point injections). The numbers from 1 to 8 represent the different individuals in the study. HV, high volume; LV, low volume.

**Table 1 animals-12-02165-t001:** The table shows the number and percentage of intercostal nerves between T2–T7 stained after performing the two-point technique for the transverse approach to the transversus thoracis plane with both volumes.

Intercostal Nerves	LV	HV	*p*-Value *
T2	3 (37.5%)	8 (100%)	0.026
T3	8 (100%)	8 (100%)	NA
T4	5 (62.5%)	8 (100%)	0.200
T5	4 (50%)	8 (100%)	0.077
T6	7 (87.5%)	8 (100%)	1
T7	1 (12.5%)	5 (62.5%)	0.119

* Fisher´s exact test. LV, low volume (0.5 mL kg^−1^); HV, high volume (1 mL kg^−1^).

**Table 2 animals-12-02165-t002:** The table shows the number and percentage of transversus thoracis plane intercostal segments between T2–T7 stained after performing the two-point technique for the transverse approach to the transversus thoracis plane with both volumes.

Region	LV	HV	*p*-Value *
TTP (2)	6 (75%)	8 (100%)	0.467
TTP (3)	8 (100%)	8 (100%)	NA
TTP (4)	5 (62.5%)	8 (100%)	0.200
TTP (5)	5 (62.5%)	8 (100%)	0.200
TTP (6)	7 (87.5%)	8 (100%)	1
TTP (7)	1 (12.5%)	5 (62.5%)	0.119

* Fisher’s exact test. HV, high volume (1 mL kg^−1^); LV, low volume (0.5 mL kg^−1^). NA, not applicable; TTP, transversus thoracis plane.

## Data Availability

Data supporting the reported results can be sent to anyone interested by contacting the corresponding author.

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
