# Peer review of "A Two-Point Ultrasound-Guided Injection Technique for the Transversus Thoracis Plane Block: A Canine Cadaveric Study"

_animals, 2022, doi:10.3390/ani12172165_

Round 1

Reviewer 1 Report

Thank you for submitting your interesting research results. My comments are as follows.

[Overal comments]

Increasing the dose and administering at two sites allowed for a high degree of staining in the T2-T7 region, but the needle should insert four times to eliminate pain in the left and right chest wall. Please describe your opinion on the advantages of this block comared to the paravertebral block.

Why you perform this block? Is it because paraspinal blocks of the cranial thoracic spine are difficult to place ultrasound probe?

[Materials and Methods]

I interpreted that in the second phase of the study, low and high volume of stain were administered to the same dog on one side of the chest wall each. Please describe whether each was administered on the left or right side.

Author Response

Dear reviewer,
Thank you for your comments and questions to improve our manuscript.
We attach below a point-by-point response to your comments and questions
Best regards!

[Overal comments]

 Increasing the dose and administering at two sites allowed for a high degree of staining in the T2-T7 region, but the needle should insert four times to eliminate pain in the left and right chest wall. Please describe your opinion on the advantages of this block comared to the paravertebral block.

Why you perform this block? Is it because paraspinal blocks of the cranial thoracic spine are difficult to place ultrasound probe?

Thank you for this interesting question. This block aims to reduce the multiple injections that would be necessary, whether it is decided to perform a multiple paravertebral or intercostal block, to desensitize the sternum. In this way, we believe that the efficacy of the block is increased as there is less probability of failure during the puncture. On the other hand, selective desensitization of the ventral chest wall is aimed, avoiding the involvement of the intercostal muscles in the block and, therefore, a greater impact on the patient's ventilatory function.

[Materials and Methods]

 I interpreted that in the second phase of the study, low and high volume of stain were administered to the same dog on one side of the chest wall each. Please describe whether each was administered on the left or right side.

Thanks for your question. Yes, indeed, in each cadaver the low volume was administered in one hemithorax and the high volume in the other. To avoid bias, the hemithorax selected for each volume was randomized. We have stated it in M&M in the manuscript.

Reviewer 2 Report

Dear authors,

this is a very interesting study, well explained, with sections presented in a balanced and coherent way. Introduction is properly documented, and the data are properly discussed. I suggest some revision providing some comments:

Simple summary: you have to improve this part. For example you should better specify how your study could be valuable for the veterinary world and for the society.

Materials and Methods: Why were 10 canine cadaveric selected? Has been done an evaluation of the sample size? Why don't you say anything about the age of cadavers? could this affect the study?

Author Response

Dear reviewer,
Thank you for your comments and questions to improve our manuscript.
We attach below a point-by-point response to your comments and questions
Best regards!

Simple summary: you have to improve this part. For example you should better specify how your study could be valuable for the veterinary world and for the society.

Thank you for your suggestion. The authors have now rewritten the Simple summary.

Materials and Methods: Why were 10 canine cadaveric selected? Has been done an evaluation of the sample size? Why don't you say anything about the age of cadavers? could this affect the study?

Thank you for this comment. All canine cadavers proceed from adult animals. We have added that in the manuscript. Considering this, we do not believe that age was a factor that could have influenced this study.

Although we have not carried out the power analysis, we have based on other similar articles to select the number of animals in the study.